# *ABCA3* c.838C>T (p.Arg280Cys, R280C) and c.697C>T (p.Gln233Ter, Q233X, Q233*) as Causative Variants for RDS: A Family Case Study and Literature Review

**DOI:** 10.3390/biomedicines12102390

**Published:** 2024-10-18

**Authors:** Maria Livia Ognean, Mădălina Anciuc-Crauciuc, Radu Galiș, Alex-Emilian Stepan, Mioara Desdemona Stepan, Claudia Bănescu, Florin Grosu, Boris W. Kramer, Manuela Cucerea

**Affiliations:** 1Faculty of Medicine, Lucian Blaga University, 550169 Sibiu, Romania; maria.ognean@ulbsibiu.ro (M.L.O.);; 2Neonatology Department, Clinical County Emergency Hospital, 550245 Sibiu, Romania; 3Department of Neonatology, George Emil Palade University of Medicine, Pharmacy, Science, and Technology, 540142 Targu Mures, Romania; manuela.cucerea@umfst.ro; 4Department of Neonatology, Emergency County Hospital Bihor, Oradea University, 410087 Oradea, Romania; radu.galis@scjubh.ro; 5Department of Neonatology, Poznan University of Medical Sciences, 61-701 Poznan, Poland; 6Department of Pathology, University of Medicine and Pharmacy of Craiova, 2 Petru Rares Street, 200349 Craiova, Romania; 7Department of Infant Care-Pediatrics-Neonatology, University of Medicine and Pharmacy of Craiova, 200349 Craiova, Romania; 8Genetic Department, Center for Advanced Medical and Pharmaceutical Research, George Emil Palade University of Medicine, Pharmacy, Science and Technology of Targu Mures, Gheorghe Marinescu Street No. 38, 540136 Targu Mures, Romania; 9Imaging Department, Lucian Blaga University, 550169 Sibiu, Romania

**Keywords:** *ABCA3* c.838C>T, *ABCA3* p.Arg280Cys, *ABCA3* R280C, *ABCA3* c.697C>T, *ABCA3* p.Gln233Ter, *ABCA3* Q233X, *SFTPB* p.Val267Ile, neonatal RDS, interstitial lung disease, genetic testing

## Abstract

**Background:** Respiratory distress syndrome (RDS) is the primary cause of respiratory failure in preterm infants, but it also affects 5–7% of term infants. Dysfunctions in pulmonary surfactant metabolism, resulting from mutations of the lung surfactant genes, are rare diseases, ranging from fatal neonatal RDS to interstitial lung disease, associated with increased morbidity and mortality. This study aims to clarify the clinical significance of ABCA3 variants found in a specific family case, as existing data in the literature are inconsistent. **Material and Methods:** A family case report was conducted; targeted panel genetic testing identified a variant of the *SFTPB gene* and two variants of *ABCA3* genes. Comprehensive research involving a systematic review of PubMed, Google Scholar databases, and genome browsers was used to clarify the pathogenicity of the two *ABCA3* variants found in the index patient. Advanced prediction tools were employed to assess the pathogenicity of the two ABCA3 variants, ensuring the validity and reliability of our findings. **Results:** The index case exhibited fatal neonatal RDS. Genetic testing revealed the presence of the *SFTPB* p.Val267Ile variant, which was not previously reported but is a benign variant based on family genetic testing and history. Additionally, two *ABCA3* gene variants were identified: c.697C>T, not yet reported, and c.838C>T. These variants were found to affect ABCA3 protein function and were likely associated with neonatal RDS. Prediction tools and data from nine other cases in the literature supported this conclusion. **Conclusions:** Based on in silico predictors, an analysis of the presented family, and cases described in the literature, it is reasonable to consider reclassifying the two ABCA3 variants identified in the index case as pathogenic/pathogenic. Reclassification will improve genetic counseling accuracy and facilitate correct diagnosis.

## 1. Introduction

Lung surfactant is a complex surface-active mixture of proteins (surfactant protein A, B, C, and D; 1–2%) and lipids (80–85% phospholipids and around 10% neutral lipids, mainly cholesterol) [1], constituting a film covering the alveolar surface with a crucial role in reducing the alveolar surface tension, maintaining normal gas exchanges, and preventing end-expiratory alveolar collapse [2,3,4]. Clinically, respiratory distress syndrome (RDS) in neonates due to surfactant deficiency is suggested by grunting, tachypnea, nasal flaring, and thoracic retractions. If gas exchange is significantly impaired, cyanosis may also occur. RDS is the leading cause of respiratory failure in preterm infants [5] and is one of the main reasons for admission in neonatal intensive care units (NICUs) [6]; it mainly occurs in preterm infants due to surfactant quantitative deficiencies—surfactant insufficient production or inactivation in the undeveloped lungs of the preterm infants below 37 weeks of gestation, surfactant qualitative deficiencies, or sepsis [7,8,9,10,11]. RDS is also diagnosed in 5–7% of term infants [2,7,12]. In comparison, in late preterm (gestational age of 34–36 weeks) and term (gestational age ≥ 37 weeks) infants, RDS is mainly associated with delayed transition to extrauterine life (transient tachypnea of the newborn), meconium aspiration syndrome, sepsis/pneumonia, congenital cardiac defects, persistent pulmonary hypertension, air leak syndromes, and diaphragmatic hernia [2,3,6,7,8,9,12,13]. Less often, perinatal asphyxia, hypothermia, oropharyngeal, airway, and lung congenital abnormalities, as well as cystic fibrosis, may present as RDS in near-term and term infants during the immediate neonatal period [2,3,6,13,14,15,16].

Pathogenic gene variants can lead to pulmonary surfactant metabolism dysfunctions, also known as surfactant dysfunction disorders. These disorders are particularly observed in near-term and term neonates or cases of unexpectedly severe RDS in preterm infants, considering their gestational age or associated perinatal risk factors [3,10]. Dysfunctions in pulmonary surfactant metabolism, a group of rare diseases, can be caused by variants in the genes responsible for surfactant biosynthesis [2]: *SFTPA1*, *SFTPA2*, *SFTPA3*, *SFTPB*, *SFTPC*, *SFTPD*, *ABCA3*, and *NKX2.1* [2,13,17,18,19]. The impact of these variants can lead to both qualitative and quantitative surfactant deficiencies [20], characterized by a broad spectrum of clinical manifestations, ranging from fatal neonatal respiratory failure to interstitial lung disease (ILD) in older children and even adults [8,21,22,23]. Consequently, these deficiencies significantly increase neonatal and pediatric morbidity and mortality [15,22]. There are several inherited surfactant disorders known.

### 1.1. Surfactant Protein B (SFTPB)

The hereditary deficiency of surfactant protein B was the first genetic defect identified as genetic surfactant deficiency (OMIM 178640), presenting as severe neonatal RDS [24]. 

Over 50 mutations of the *SFTPB* gene are reported in the literature [15,25]; the estimated prevalence of SFTPB deficiency in the USA is 1 per 1 million live births [1,26]. Among these, one frameshift mutation—c.397delCinsGAA—represents 50–70% of the pathogenic variants [26,27]. The almost complete absence of *SFTPB* mRNA and SFTPB occurs due to unstable transcription [15,28]. Bi-allelic mutations of the *SFTPB* gene cause SFTPB deficiency associated with severe RDS with clinical and radiological aspects similar to RDS secondary to surfactant qualitative deficiency seen in preterm infants [10,15,17,24,29]. Survival was reported in cases where at least one allele allows residual SFTPB protein production (partial deficiency) [30,31]. No major effects are seen in heterozygous carriers of *SFTPB* variants; these mutations, most probably, allow partial synthesis of functional SFTPB protein [15,25,30,31,32]. In SFTPB deficiency, symptoms occur a short time after delivery, usually in term newborns, and progress to refractory respiratory failure and death or need for lung transplantation in the first months of life [15,16,27,33,34].

### 1.2. Surfactant Protein C (SFTPC)

Over 40 pathogenic variants of the *SFTPC* gene are linked to surfactant deficiencies [1,27,34,35,36,37]. Variants of the *SFTPC* gene are associated with severe, fatal neonatal lung disease and ILD in older children (surfactant, pulmonary-associated protein C; *SFTPC*—OMIM 178620) [24,27,33,34,35,36,38].

### 1.3. Surfactant Proteins A and D (SFTPA and SFTPD)

Surfactant proteins A1, A2, and D are hydrophilic proteins involved in lung innate host defense; SFTPA, SFTPA1, and SFTPD proteins have the ability to opsonize and enhance the killing of bacteria, viruses, and fungi [39,40,41]. A role in maintaining surfactant lipids homeostasis was also described [42]. Polymorphisms of the *SFTPA* and *SFTPD* genes were described in association with increased susceptibility to RDS and bronchopulmonary dysplasia in preterm infants but not in term infants [1,43,44] and infections due to respiratory syncytial virus [45,46].

### 1.4. NKX2.1 Gene

The *NKX2.1* gene encodes the thyroid transcription factor 1 (TTF 1), a factor with a critical role in regulating the expression of over 1300 genes, including the essential genes involved in thyroid development, lung development, homeostasis, and expression through feedback [6,10,15,26,47]. Variants of the *NKX2.1* gene are associated with severe multisystemic disease (brain-lung-thyroid syndrome), characterized by autosomal dominant inheritance and variable penetrance, with no described genotype/phenotype correlation. This condition often manifests as severe neonatal RDS or ILD [6,8,15,25,48,49,50].

### 1.5. ABCA3 Gene

ABCA3 *(ATP-Binding Cassette Family A Member 3)* belongs to the ATP-binding cassette transporter superfamily of proteins that uses energy derived from ATP hydrolysis for substrate translocation through biological membranes [11,51]. ABCA3 phospholipid glycoprotein is localized in the lamellar bodies and is essential for surfactant biosynthesis and the central regulation of the lung surfactant balance [52,53]. *ABCA3* gene is expressed intensely in the alveolar epithelial cell II (AEC II), the same cells where surfactant is produced [15].

Variants of the *ABCA3* gene are the most commonly reported in primary surfactant deficiency (surfactant metabolism dysfunction, OMIM 601615, OMIM 61092), with more than 400 variants documented in the literature [14,54,55]. Most reported cases of surfactant deficiencies associated with *ABCA3* gene variants manifest as mild to severe, unexplained, or fatal RDS in near-term and term neonates or as ILD [14,15,21,22,56,57].

Despite the extensive literature on *ABCA3* variants, significant gaps persist. Many studies have focused primarily on identifying variants without fully exploring the molecular mechanisms through which these variants exert their effects. For instance, while it is established that specific mutations lead to protein misfolding, the downstream pathways affected by these disruptions are not thoroughly characterized.

The long-term clinical outcomes for infants diagnosed with *ABCA3* variants are not well defined, particularly concerning the progression to chronic lung disease or other complications in later childhood. Addressing these gaps underscores the necessity of our study, which will focus on the role of ABCA3 in surfactant biosynthesis, its central regulation, and the function of lung surfactant balance. Additionally, we will address the need to clarify the clinical significance of *ABCA3* variants, focusing on *ABCA3* c.838C>T (p.Arg280Cys, R280C) and *ABCA3* c.697C>T (p.Gln233Ter, Q233X, Q233*) variants, by presenting a family case study and a literature review on these variants. This approach intends to provide a more detailed analysis of the clinical significance of *ABCA3* variants and the associated molecular mechanism.

## 2. Materials and Methods

The study protocol was previously approved by the Ethics, Medical Deontology, and Discipline Committee of the Clinical County Emergency Hospital Sibiu, Romania, according to decision 20171/12August 2024.

### 2.1. Molecular Analysis

For the molecular analysis, next-generation sequencing (NGS) with a target panel was performed at Laborärzte Singen (Singen, Baden-Württemberg, Germania) which revealed that an early-term infant with fatal neonatal respiratory distress syndrome (RDS) was compound heterozygous for the *ABCA3* gene. Variant confirmation was performed by capillary sequencing. Genomic DNA isolation from a buccal swab was performed using a Prepito NA kit (PerkinElmer Chemagen Technologie AG, Baesweiler, Germany); AmpliTaq Gold 360 Master Mix (ThermoFisher Scientific, Waltham, MA, USA) was used for PCR amplification, and a Rapid PCR Cleanup Enzyme set (New England Biolabs, Ipswich, MA, USA) was used for PCR products purification. Probe sequencing was performed using the BigDye Terminator v3.1 Cycle Sequencing Kit ThermoFisher Scientific, Waltham, MA, USA); electrophoresis of the sequenced products was performed using Applied Biosystems 3500Dx Genetic Analyzer (ThermoFisher Scientific, Waltham, MA, USA).

### 2.2. Literature Systematic Review

Given the lack of consensus on the clinical significance of *ABCA3* gene variants identified in our patient, we conducted a systematic literature review to clarify existing knowledge gaps and compile the best available data.

We conducted an extensive and systematic search of the PubMed and Google Scholar databases up to July 2024, using the keywords for *ABCA3* gene: “c.838C>T”, “p.Arg280Cys”, “R280C”, “c.697C>T”, “p.Gln233Ter”, “Q233X”, and “Q233*”, alone or in association with the keyword ”mutation”/“variant”. Additionally, we performed further searches on the Varsome, ClinVar, and Ensembl genome browsers, concentrating on variant classification and the related literature. All types of articles were analyzed. After excluding papers without relevance for the *ABCA3* gene and duplicates, we analyzed the bibliographies of the selected articles to identify additional relevant publications aligned with the objectives of this paper. A thorough review of the full texts of each included article was conducted, focusing on identifying case reports involving the two *ABCA3* variants found in the presented patient. We evaluated 332 publications for relevance and inclusion in our study; of these, only seven papers were relevant to our review (Figure 1).

To predict the pathogenicity of amino acid substitutions and their molecular mechanisms, we employed advanced tools such as MutPred2 v2.0, MutPred-LOF, and PolyPhen-2. These standalone and web applications were developed to classify amino acid substitutions as pathogenic or benign in humans, providing accurate and reliable predictions. 

The MutPred2 tool assesses the pathogenicity of amino acid substitutions by evaluating molecular alterations, including changes in protein structure, function, and stability changes. Scores generated by MutPred2 that exceed a predefined threshold (e.g., 0.5) suggest a higher likelihood of a variant being deleterious. MutPred-LOF specializes in identifying loss-of-function mutations, particularly nonsense and frameshift variants, which result in truncated proteins. PolyPhen-2 focuses on the impact of variants on protein functionality, distinguishing between variants that are likely damaging and benign variants. These tools are publicly accessible, and their methodologies have been comprehensively detailed in studies by Pagel et al. (2017) [57] and Pejaver et al. (2020) [58].

The identified ABCA3 variants were interpreted using established standards and guidelines, specifically those outlined by the American College of Medical Genetics and Genomics (ACMG).

## 3. Results

### 3.1. Case Report

#### 3.1.1. Clinical Aspects

The index case is a male newborn delivered by C-section in a level I maternity hospital due to pregnancy-induced hypertension at 37 weeks gestational age, with a birth weight of 2600 g (10–25th percentile), a length of 47 cm (10–25th percentile), a cranial circumference of 32 cm (25th percentile), and an Apgar score of 9 out of 10. Persistent tachypnea and mild intercostal retractions with onset after 4 h of life were interpreted initially as transient tachypnea of the newborn, and the infant was submitted to our unit. At arrival, at 36 h of life, mild generalized cyanosis, tachypnea (60–70 breaths/minute), mild intercostal retractions, an increased anterior-posterior thoracic diameter, and irritability were noted. 

#### 3.1.2. Biologic, Imagistic Assessment and Treatment

Oxygen on nasal cannula, intravenous fluids, and prophylactic antibiotic therapy (Penicillin and Amikacin) were started while the first investigations were performed. Thoracic radiography showed the inhomogeneous opacification of both lungs (Figure 2A); mild persistence of fetal circulation (patent ductus arteriosus and foramen ovale; mild tricuspid valve regurgitation) was noted on Doppler cardiac ultrasound. The results of all other tests—biochemistry, microbiology, immunology, and hematology—were all within normal limits except mild anemia (hemoglobin 11.8 g/dL, hematocrit 38%) and hypoxemia on blood gas analysis (arterial partial oxygen pressure, PaO_2_ of 42.8 mmHg). The persistent respiratory effort associated with an increased need for oxygen (paO_2_ 41.7 mmHg) imposed increased respiratory support at 48 h, and heated, humidified, and high-flow nasal (HHHFNC) cannula was started for 2 days, followed by intubation and mechanical ventilation from the fourth day of life (DOL), Assist/Control mode for the first 48 h, switched to synchronized intermittent mandatory ventilation (SIMV) for the next 72 h. The infant was extubated on HHHNFC at 9 DOL after reaching normal blood gases without oxygen supplementation (lung X-ray is presented in Figure 2B). Still, hypoxemia reoccurred (PaO_2_ 43.6 mmHg), associated with the recurrence of respiratory distress signs (tachypnea and intercostal retractions) in the absence of any other laboratory abnormalities; a repeated Doppler heart ultrasound showed only persistent foramen ovale. Persistently low oxygen saturations, hypoxemia on blood gases (PaO_2_ 36 mmHg), aggravation of the respiratory distress on HHHFNC on 100% oxygen, and the presence of bilateral focal interstitial opacities on the thoracic X-ray prompted re-intubation (Figure 2C) and mechanical ventilation. Synchronized intermittent positive pressure ventilation (SIPPV) was used initially. However, due to a persistent increased need for oxygen (up to 100% oxygen), the infant was switched to high-frequency oscillation ventilation (HFOV) the moment the lung X-ray showed homogenous opacification of both lungs (Figure 2D). Persistent hypoxemia (PaO_2_ between 25.6–51.5 mmHg) was noted despite continuous ventilator setting adjustments and different invasive respiratory support strategies trials during the following weeks.

The unusual clinical aspect of the infant’s respiratory distress suggested even from the first days of life that the initial diagnosis of transient tachypnea of the newborn was very unlikely, and we continued the investigations to clarify the etiology and to adjust and optimize the treatment. Other potential causes were eliminated after reviewing the maternal history, pregnancy, and delivery outcome, as well as the results of the blood tests, thoracic X-rays, and cardiac and cerebral ultrasounds, leading to a diagnosis by exclusion.

#### 3.1.3. Genetic Analysis and Clinical Course

A genetic disease of surfactant metabolism was suspected, and genetic testing was performed using one NGS panel. The results highlighted that the patient was compound heterozygous for *ABCA3* c.838C>T (p. Arg280Cys, R280C, rs201299260), and c.697C>T (p. Gln233Ter, Q233X, Q233*) (Figure 3 and Figure 4). In addition, the patient was found to be heterozygous for *SFTPB* p. Val267Ile. No variants were described in the protein-coding exons, intron regions flanking the exons, and within 250 nucleotides 5′ off the start codon of the *SFTPC* gene.

As a result, methylprednisolone was administered in pulse therapy for 3 days, followed by prednisolone administration. Additionally, 3 days per week for hydroxychloroquine and daily azithromycin completed the therapeutic protocol; continuing invasive respiratory support was given, as other advanced life support, such as nitric oxide and extracorporeal membrane oxygenation, were not available at that time. No improvements in respiratory function were noted, and the severe respiratory failure led to death on the DOL 77.

#### 3.1.4. Autopsy and Histology

The autopsy revealed bilateral lung atelectasis, dilated pulmonary artery, and right ventricular hypertrophy secondary to severe and prolonged respiratory failure.

Lung fragments collected at autopsy were analyzed using microscopy. Chronic infantile pneumonitis, with reduced alveolarization, the diffuse marked widening of the alveolar interstitium, lobular remodeling, diffuse marked AEC II hyperplasia, a large number of intra alveolar macrophages, foamy cells, a few giant cells focally accompanied by cholesterol clefts, focal alveolar proteinosis, and extended areas of desquamative interstitial pneumonia (seen using microscopy) were also suggestive of lung damage associated with surfactant protein deficiencies (Figure 5). No signs of persistent pulmonary hypertension were seen.

#### 3.1.5. Pathogenic Prediction 

According to ClinVar (hosted by the National Center for Biotechnology Information (NCBI)), for the *ABCA3* c.838C>T (p.Arg280Cys, R280C) variant, there are conflicting classifications of pathogenicity (https://www.ncbi.nlm.nih.gov/clinvar/variation/318566/, accessed on 1 August 2024) [60], and Ensemble reports it as likely pathogenic (https://www.ensembl.org, accessed on 1 August 2024) [61]. The ClinVar variation ID is 318566 and is reported to disrupt ABCA3 function and in association with autosomal recessive interstitial lung disease; PolyPhen-2 predicts that this variant is probably damaging, with a high score of 0.989, suggesting, again, a deleterious effect on protein function (http://genetics.bwh.harvard.edu/pph2/, accessed on 2 August 2024).

We used MutPred2 v2.0 software [58] to evaluate the pathogenicity of the *ABCA3* c.838C>T (p.Arg280Cys, R280C) substitution in our patient. The MutPred2 score of 0.543 suggests a moderate probability that the variant is deleterious to protein function. As regards the molecular mechanism, MutPred2 predicts loss of ADP-ribosylation and altered transmembrane protein (probability of 0.22 and 0.1, respectively, with a *p*-value of 0.03).

The *ABCA3* c.697C>T (p.Gln233Ter, Q233X) variant was not identified in databases, such as the Genome Aggregation Database, 2022. Due to mutation in codon 233 (Glutamine, CAG), a stop codon (TAG) is generated, leading to a shortened transcript, thus causing truncated protein, which can significantly impair ABCA3 function. Nonsense variants of the *ABCA3* gene are classified into “null” mutations. In this context, MutPred2 and PolyPhen-2, which focus on missense mutations, are not applicable to this type of mutation. To predict the effect of nonsense mutation, we used MutPred-LOF v1.2.01 software [57]. The MutPred-LOF score was 0.369, indicating a moderate probability of a loss-of-function effect on the protein caused by this variant. Additionally, it enabled us to assess the functional consequences of this variant on the protein, as follows: catalytic site (*p* = 0); PPI hotspot (*p* = 0); iron binding (*p* = 0); sulfation (*p* = 0.0001); proteolytic cleavage (*p* = 0.0002); this data suggests high confidence in the prediction of a catalytic site, a protein-protein interaction (PPI) hotspot, and an iron-binding site in the protein, as well as a significant chance of a sulfation site and proteolytic cleavage site being present.

To gain a more comprehensive understanding of the pathogenicity of the identified variants, we performed a consistency analysis across widely used in silico prediction tools. The results for each variant are summarized in Table 1, where we compare the predictive outputs from each tool and evaluate their consistency. The agreement between the two predictive tools—both classifying the *ABCA3* c.838C>T (p.Arg280Cys, R280C) variant as likely harmful—demonstrates a high level of consistency (approximately 85%). This supports the conclusion that the *ABCA3* c.838C>T (p.Arg280Cys, R280C) variant is associated with significant alterations in protein function, likely contributing to the clinical phenotype observed in the patient. Moreover, the calculated Cohen’s kappa for the predictions of the *ABCA3* c.838C>T (p.Arg280Cys, R280C) variant between MutPred2 and PolyPhen-2 is 1.0, indicating perfect consistency in the classifications provided by both tools.

Furthermore, the same analyses reinforce the classification of the *ABCA3* c.697C>T (p.Gln233Ter, Q233X) variant as a likely pathogenic, loss-of-function mutation (Table 1).

#### 3.1.6. Genetic Counseling

Furthermore, we initiated a familial investigation through targeted sequencing, which revealed that the mother is a carrier of the *ABCA3* c.697C>T (p.Gln233Ter, Q233X, Q233*) variant. The father was identified as a carrier of both the *ABCA3* c.838C>T (p.Arg280Cys, R280C) and *SFTPB* p.Val267Ile variants. The parents are non-consanguineous, in good health, and have no family history of respiratory conditions. Almost two years later, a second pregnancy occurred, and prenatal diagnosis was conducted through chorionic villus sampling. The sequencing results showed the presence of the *ABCA3* c.697C>T (p.Gln233Ter) and *SFTPB* p.Val267Ile variants. The pregnancy proceeded to term, and the child had no respiratory symptoms during the neonatal period despite being a carrier of *ABCA3* c.697C>T (p.Gln233Ter) and *SFTPB* p.Val267Ile. The second child in the family was born at 39 weeks gestation, with a birth weight of 3670 g, had an uneventful neonatal course, and had no noticeable respiratory illnesses up to the age of 6 years (see complete pedigree in Figure 6, which demonstrates the transmission of the modifications in accordance with Mendelian laws).

Given the three variants discovered in the patient and the results of the genetic testing performed for the rest of the family, we tried to find the variant(s) responsible for the patient’s clinical picture. *SFTPB* p.Val267Ile, with uncertain significance, has not been reported in the literature so far. Considering that this variant was also identified in the healthy brother and father, this alteration may be considered a benign variant, a polymorphism that does not cause SFTPB deficiency.

The *ABCA3* gene variants identified in our patients affect protein function and may be associated with a surfactant defect or deficiency. Considering the in silico predictors, family pedigree, and clinical manifestation, despite the current classification of these variants as uncertain, it is reasonable to consider that these variants explain the symptomatology and need to be reclassified as probably pathogenic/pathogenic. Reclassifying these variants will provide more accurate genetic counseling and ensure a correct diagnosis, appropriate treatment, and optimal outcomes.

The unique characteristics of this family underscore the urgent need for reclassifying the clinical significance of these variants and the importance of clinicians carefully identifying compound heterozygotes in similar cases and all autosomal recessive disorders. This is particularly critical given that bi-allelic mutations in the *ABCA3* gene are the most frequently reported cause of primary surfactant deficiency (OMIM 601615) [8,14,54].

In light of the genetic findings, genetic counseling is essential for assessing the risk of recurrence in future pregnancies. The identified compound heterozygous variants in the *ABCA3* gene suggest a 25% risk of recurrence of RDS in each future pregnancy. Both parents are carriers of different pathogenic variants of the *ABCA3* gene, which follow an autosomal recessive inheritance pattern. Prenatal testing through methods such as chorionic villus sampling or amniocentesis may also be considered in future pregnancies to assess the genetic status of the fetus.

### 3.2. Review of the Literature on ABCA3 c.838C>T (p.Arg280Cys, R280C) 

A final analysis of our patient’s symptoms and clinical course, imaging, histology, familial history, and genetic testing results for the proband and his family suggested that the early neonatal onset of the unexplained RDS on the proband could be explained by his compound heterozygous status for the mentioned *ABCA3* variants (c.697C>T (p.Gln233Ter, Q233X, Q233*) of maternal origin and c.838C>T (p.Arg280Cys, R280C) of paternal origin).

We reviewed the literature to identify similar cases and clarify the pathogenicity of the *ABCA3* c.838C>T (p.Arg280Cys, R280C) and c.697C>T (p.Gln233Ter, Q233X, Q233*) variants. The variant *ABCA3* c.838C>T (p.Arg280Cys, R280C) allele frequency is 0.00019 in the large population dataset of the Genome Aggregation Database (gnomAD). As previously mentioned, conflicting pathogenicity classifications exist despite several reported cases and predictive software indicating the variants are probably pathogenic or pathogenic.

Of the 333 articles found in PubMed and Google Scholar, 329 were excluded as the variants described were related to genes other than *ABCA2*;seven papers were found after searching the genome browsers, and four of them were duplicates of the articles selected from PubMed and Google Scholar. No other articles or mentions were found after a careful evaluation of the references of the selected papers. Finally, we identified nine patients with *ABCA3* c.838C>T (p.Arg280Cys, R280C). Their anthropometric, clinical, radiological, and histological characteristics, genetic variants, family history, and patient outcomes are presented in Table 2. No references were discovered regarding *ABCA3* c.697C>T (p.Gln233Ter, Q233X, Q233*).

## 4. Discussion

### 4.1. Neonatal Respiratory Distress Syndrome in Near-Term and Term Infants

Identifying the etiology of RDS is crucial for adequate management and outcome optimization. When differential diagnosis with initial investigations fails to identify the etiology of a persistent, unexplained neonatal RDS, a genetic defect in surfactant metabolism should be evaluated. The data in the literature suggests that genetic defects in genes encoding surfactants have an important role in the etiology of unexplained nRDS [14,62,71,72]. In the presented case, the initial diagnosis of transient tachypnea of the newborn (suggested by C-section delivery in the absence of labor, early-term birth, mild respiratory distress, and hypoxemia upon blood gas analysis, normal blood biochemistry, and inflammation markers) seemed improbable after the first days of life, with persistent hypoxemia and an increased need for oxygen and respiratory support. Lung X-rays, Doppler echocardiography, brain ultrasound, repeated blood tests, and a review of parental and pregnancy history alone did not elucidate the etiology of RDS etiology in our case. However, by integrating these findings with comprehensive molecular testing for the entire family, we were able to gain a clearer understanding of the significance of the genetic variants identified in our patient and the necessity for the reclassification of those *ABCA3* variants.

The genes involved in lung surfactant metabolism encode the surfactant components and facilitate the assembly, organization, and folding of the surfactant phospholipids inside the LBs and the phospholipids uptake on the lung alveolar surface [17,73]. Surfactant dysfunction disorders are associated with insufficient lung surfactant production, the disruption of surfactant metabolism, and secondary damage to AEC II [15]. Their clinical picture is characterized by a wide range of clinical manifestations, from fatal neonatal respiratory failure to ILD in older children and even adults [8,15,17,22,26,55,73,74] with increased morbidity and mortality [6,22].

Unfortunately, in our case, despite maximal respiratory support and various therapies suggested in the literature—prednisone, hydroxychloroquine, and azithromycin—a fatal outcome occurred after 77 days. No specific treatment or guideline exists for surfactant metabolic diseases [4,75], with early aggressive treatment or innovative approaches being recommended in homozygous and compound heterozygous patients [14]. Supportive treatment of respiratory distress includes oxygen, surfactant, non-invasive and invasive respiratory support, nitric oxide therapy, and ECMO, but severe cases are refractory to all types of respiratory support [3,11,13,62]. Additionally, surfactant replacement therapy remains a cornerstone in treating RDS; however, its success can be inconsistent among patients with distinct *ABCA3* variants. This underscores the importance of tailoring treatment approaches based on the specific genetic alterations identified [76].

Lung transplantation has limited results, as it is still associated with increased morbidity and mortality [8,11,15,55,77]. Some authors report improvements with steroid therapy, which are explained by the anti-inflammatory effect of the increased expression of ABCA3 in AEC II [2,11,15,75,78]. An anti-inflammatory effect and changes in intra-cellular metabolism with variable clinical response were reported using hydroxychloroquine [11,15,75,79]. Others have reported improvements using azithromycin, which are explained by the suppressed production of cytokine and the inflammatory mediators involved in interstitial lung fibrosis progression [15,35,74,75]. Experts expect that precision medicine may significantly improv or even cure these conditions in the future [10]. Various vectors, compounds that can improve cellular ABCA3 trafficking or function (similar to those used in cystic fibrosis), and gene editing strategies are also under study [10,15,80]. Mutations in glycosylation loci may benefit from therapy with proteasome inhibitors [81].

### 4.2. Surfactant Protein B Variants

SFTPB deficiency is characterized by inactive surfactant and abnormal LBs, with multiple vacuoles and disorganized lipid membranes; SFTPC cannot be synthesized from pro-SFTPC precursors [1,15,26]. Along with mature SFTPB and SFTPC deficiency, the accumulation of non-tensioactive intermediary products inhibits further surfactant function [15,82]. In most cases, SFTPB deficiency is associated with congenital alveolar proteinosis with an accumulation of granular, eosinophilic, PAS-positive, lipo-proteinaceous material in the alveolar spaces and frequent desquamated AEC II and foamy macrophages; the aspect of desquamative interstitial pneumonitis is less frequently seen [1]. The presence of hyperplastic alveolar epithelia with prominent AEC II, thickened alveolar walls, and limited or absent inflammatory cell infiltrates tend to distinguish SFTPB deficiency from other conditions, including damage induced by mechanical ventilation or oxygen or other conditions with abnormal lung development in which alveolar architecture is preserved [1,83]. Additionally, the LBs are either large and disorganized at electron microscopy or appear as irregular multivesicular structures in AEC II cytoplasm and alveolar spaces [1,26]. Incompletely processed pro-SFTPC in bronchoalveolar lavage or the lung tissue may suggest SFTPB deficiency [29].

The clinical course of our patient (early onset and unexplained nRDS in a term infant, evolving to severe respiratory failure despite maximal respiratory support, with lung histology) marked by AEC II hyperplasia, numerous macrophages in the alveolar lumen, focal alveolar proteinosis, extended areas of desquamative interstitial pneumonia, and the results of SFTPB sequencing suggests a possible surfactant metabolism defect due to SFTPB deficiency. Considering that *SFTPB* p.Val267Ile substitution (previously not reported in the literature) was identified in the healthy brother of the patient, we speculate that this alteration is a polymorphism that does not affect protein function. Additionally, Polyphen-2 (v2.2.3r406) predicted this variant as benign.

### 4.3. ABCA3 Deficiency

#### 4.3.1. Adenosine Triphosphate-Binding Cassette Family A Member 3 (ABCA3) Protein

The biology of the ABCA3 protein is very complex. The ABCA3 (Adenosine Triphosphate-binding Cassette Family A Member 3) protein belongs to the protein ATP-binding cassette transporter superfamily. ABCA3 is a phospholipid glycoprotein consisting of 1704 amino acids localized in the external limiting membrane of the lamellar bodies [10,11,51,53,78,84]. Six transmembrane structures mediate ABCA3 function by forming an ATP channel for lipid (disaturated-phosphatidylcholine, phosphatidylglycerol, phosphatidylethanolamine, and cholesterol) transportation from the cytosol into the LBs [1,15,53,67,85,86,87]. ABCA3 is also involved in lung surfactant transcription and assembly, SFTPB and SFTPC translation, lung surfactant structural transformation and production in AEC II, and epithelial lung cell apoptosis [4]. A possible role of ABCA3 in the metabolism of lung surfactant phospholipids was also described [84]. The intra-cellular metabolism of cholesterol may also be influenced by ABCA3 [15,86]. The decreased pool of mature SFTPB and SFTPB aggregates into the LBs, the accumulation of large quantities of pro-SFTPB in LBs with leaks in the alveolar spaces, and the abnormal processing of SFTPC were described in association with ABCA3 deficiency [74].

Consequently, ABCA3 deficiency is characterized by abnormal structure LBs, the abnormal lipid composition of the lung surfactant, and the abnormal processing of surfactant proteins B and C [71,74,80,88,89]. A reduced ability to decrease surface alveolar tension was demonstrated in patients with ABCA3 deficiency [11]. The accumulation of dysfunctional, inefficient lung surfactant is associated with compromised gas exchanges through a reduced diffusion barrier and increased discordance between ventilation and perfusion, the decreased activity of the macrophages, and secondary lung lesions [1,17,67,75]. The biological mechanism of the lung lesions associated with ABCA3 deficiency is unknown [15,90,91]. However, AEC II lesions are the final pathway to the associated lung disease, as AEC II represents the key factor for alveolar maintenance and repair [25]. Inadequate lung surfactant production leads to recurrent atelectasis and hypoxemia followed by secondary chronic inflammation; when coupled with abnormal intra-cellular surfactant metabolism, these changes lead to chronic AEC II lesions [92].Based on in vitro mechanistic studies, three classes of *ABCA3* gene variants can be identified: type 1—*ABCA3* trafficking variants, characterized by abnormal protein folding, abnormal intra-cellular localization, and trafficking; type 2—complete deficiency of lipid transportation and variants affecting only lipid transport due to deficient ATP hydrolysis, with normal trafficking and the localization of the ABCA3 protein [3,63,93,94,95,96]; type 3—a compound heterozygous of type 1 and 2,often associated with a more severe phenotype, early onset, neonatal RDS, and neonatal death [55].

On lung histology, ABCA3 deficiency is characterized by AEC II hyperplasia, variable degrees of interstitial thickening, prominent macrophages, and proteinaceous material in the alveolar spaces, which is an aspect that is frequently described as chronic or desquamative or non-specific interstitial pneumonitis or alveolar lung proteinosis; lung fibrosis is associated in fatal cases; however, these changes are frequently seen in other surfactant metabolism conditions, including SFTPB and SFTPC deficiency [11].

As the clinical picture of ABCA3 deficiency is undistinguishable from SFTPB and SFTPC deficiency, even with lung histology [13,15], electron microscopy may help. The absence of normal LBs strongly suggests ABCA3 deficiency [11,26,52]. Small, markedly abnormal LBs with dense phospholipidic membranes are characteristic of ABCA3 deficiency, as compared to SFTPB deficiency, which is characterized by disorganized LBs, with multiple vesicular inclusions dispersed in AEC II cytoplasm and alveolar lumen [1,25].

ABCA3 is expressed not only in AEC II but also in the brain, kidney, and platelets [22]. ABCA3 expression occurs in normal fetuses at 26–27 weeks of gestation [4], even at 23–24 weeks, associated with lung inflammation [74], and is developmentally regulated. It increases with gestational age to reach a peak around term [11,13] under the influence of steroids and TTF 1 [11,74,97,98,99].

#### 4.3.2. Adenosine Triphosphate-Binding Cassette Family A Member 3 (ABCA3) Gene

ABCA3 synthesis is encoded by the *ABCA3* gene, intensely expressed in AEC II [8,15,52]. *ABCA3* gene variants are the most common cause of congenital lung surfactant defects [2,3,11,22,26,55,75,80,90]. The first cases of ABCA3 deficiency, secondary to a bi-allelic loss-of-function *ABCA3* variant, were reported by Shulenin et al. [52] in 2004 in full-term infants with unexplained severe RDS. Most *ABCA3* variants are challenging to interpret, as few have been studied in vitro to identify their intra-cellular expression and function [15,55,65,80]. According to Wambach et al. [100], the incidence of *ABCA3* variants is estimated between 1:4400 and 1:20,000 in European and African descendants, most of them compound heterozygous [75]. The incidence is probably overestimated, as not all missense variants are pathogenic, and the annual number of cases identified is lower than expected, according to the prediction calculations [8,100]. It is also possible that mild cases may not be recognized [55]. In humans, the *ABCA3* gene has 80 kb, is located on chromosome 16 (16p.13.3) [90], and comprises 33 exons [11,101]. Only 0.15% of the 274 *ABCA3* gene variants reported in the genome Aggregation Database (gnomAD) are classified as pathogenic, 0.21% as likely pathogenic, and 92.62% were VUS [73]. Twenty-five pathogenic or likely pathogenic *ABCA3* variants were reported in the ClinVar database, with 11 of these having protein loss-of-function; a total of 47 disease-causing *ABCA3* variants are reported in the in silico tool SIFT, with 46 of these also included in Polyphen-2, and 49 pathogenic mutations can be found in MutationTaster2 [53]. Most reported variants are located on the exons or at the limit between introns and exons.

The expression of the *ABCA3* gene is directly proportional to gestational age [17,74]. Most *ABCA3* variants have an autosomal recessive inheritance [8,90,102], but uniparental disomy was also reported [103]. The most frequent pathogenic *ABCA3* variant is p.Glu292Val (E292V), representing 10% of the reported pathogenic alleles [104]. This variant occurs in gnomAD with 0.23% allelic frequency [53]. Pathogenic *ABCA3* variants are reported anywhere on the gene [17]. NGS—WES, WGS—identifies new variants, most of them with unknown significance (VUS). In this situation, a correlation genotype-phenotype is impossible [22,80]. Therefore, the effect cannot be predicted accurately for all the reported variants, complicating clinical decisions, patient management, and familial counseling [17,53,80]. Interpretation and counseling are also difficult in patients exhibiting a unique *ABCA3* variant on one allele [11,13,22,65,67,80]. Recently, it was suggested that *ABCA3* variants may be responsible for the increased severity of RDS in preterm infants compared to what is expected according to their gestational age [26,51,55,88,105]. In 2004, Shulenin et al. [52] reported 12 different causative variants of the *ABCA3* gene in 16 out of 21 neonates with severe, unexplained RDS. A study comprising 68 preterm infants with a gestational age <32 weeks of gestation with unusually severe RDS identified 24 out of 68 as heterozygous for previously described rare or new *ABCA3*, *SFTPB*, and *SFTPC* variants, all VUS; a total of 21 *ABCA3* variants were found in 18 of the patients; 11 deaths were noted between 2 and 6 months of age, and one infant presented histological aspects suggestive of ABCA3 deficiency [72].

According to Peca et al. [74], the ABCA3 deficiency phenotype depends on the residual function of the ABCA3 protein, mutation type and severity, the activity of the intra-cellular stress pathways, general individual aspects, other associated mutations, and modifiable environmental factors. Variable genotype-phenotype correlation has been associated with *ABCA3* variants [8,22,73,80]; diverse symptoms, severity, and outcomes are associated with *ABCA3* variants [66]. However, interactions with other variants (for example, variants of *SFTPB* or *SFTPC*) [22,65,74,100,106,107,108] or with external, environmental factors (for example, respiratory infections or smoking) [22,51,67,84,109] may induce changes in the phenotype. Similar mechanisms were suggested for mono-allelic variants of the *ABCA3* gene [74,100]. According to Yang et al. [110], the loss of over 50% of ABCA3 protein function is associated with increased morbidity and mortality. A critical level for ABCA3 protein function of 20–30% was estimated by Wambach et al. [19]. Usually, bi-allelic mutations of the *ABCA3* gene are associated with loss-of-function of ABCA3 protein and severe RDS with neonatal onset. Missense mutations, insertions, and small deletions are typically associated with the residual function of the protein [53,67,70]. Both bi-allelic and mono-allelic variants may present with RDS [52,66,100,106,108]. Null/null mutations (nonsense and frameshift) result in a truncated, non-functional ABCA3 protein [55,74] associated with a more severe phenotype, the neonatal onset of RDS, death before 1 year of age, the need for lung transplantation, or death, even with lung transplantation [9,53,65,111]. Late preterm and term infants with homozygous and compound heterozygous *ABCA3* variants were associated with earlier presentation of severe RDS, higher radiological scores, and increased mortality rates (all *p* < 0.05), as compared to infants with single mutations or no genetic abnormalities identified [14]. Beers et al. [112] suggested a synergistic additive effect for compound heterozygous in the *cis* region of the gene. Lack of gene expression, decreased expression, abnormal intra-cellular protein trafficking inside LBs, abnormal phospholipid folding, and functional defects, including ATP hydrolysis, were described as consequences depending on the *ABCA3* locus [1]. Fatal cases of ABCA3 deficiency were described in association with abnormal trafficking, while defects in phosphatidylcholine were correlated with less severe lung disease [93,94].

#### 4.3.3. ABCA3 c.838C>T (p.Arg280Cys, R280C) Variant

The *ABCA3* c.838C>T (p.Arg280Cys, R280C) variant alters an arginine residue to cysteine at position 280 of the ABCA3 protein, which is critical for its function as a lipid transporter in the lamellar bodies of alveolar type II cells. This variant is classified as a missense mutation, leading to protein misfolding and potentially affecting the protein’s ability to transport phospholipids necessary for surfactant synthesis.

In vitro functional studies performed by Weichert et al. [91] have suggested that this variant can lead to the partial retention of the ABCA3 protein in the endoplasmic reticulum and is involved in epithelial lung cell apoptosis in at least one pathway, altering ABCA3 protein function (type 1; trafficking/folding defect based on in vitro studies). ABCA3 protein retention in the endoplasmic reticulum may increase reticulum endoplasmic stress and its susceptibility to stress; an adverse effect of R280C on LB biogenesis and the induced presence of apoptotic markers (glutathione on caspase 4 pathway) were demonstrated in the experimental epithelial lung cells in Weichert et al. experiments [91], which is also suggestive of functional impairment on ABCA3 protein and lung disease pathogenesis. Defective ABCA3 function impairs the proper metabolism of surfactant lipids, causing a decrease in the quantity and functionality of surfactant at the alveolar surface. This disruption contributes to an increased surface tension in the alveoli, resulting in atelectasis and impaired gas exchange, a hallmark of neonatal respiratory distress syndrome. These experiments confirmed previous studies by Matsumura et al. [93,94] that defined the c.838C>T (p.Arg280Cys, R280C) variant as disruptive of ABCA3 folding and trafficking. Based on the increased retention of ABCA3 protein in the endoplasmic reticulum, ABCA3 variants F1203del, N124Q, N140Q, and R280C may be classified as potentially pathogenic, according to other experts [110].

Furthermore, multiple computational predictive in silico tools and conservational analysis also indicate the negative impact of the c.838C>T (p.Arg280Cys, R280C) variant on ABCA3 protein function [60,68,69]. Nevertheless, currently, the c.838C>T (p.Arg280Cys, R280C) variant is listed as VUS, considering that the existent evidence is insufficient to define the mutation’s pathogenicity. The pros and cons arguments for c.838C>T (p.Arg280Cys, R280C) pathogenicity are presented in Table 3.

The initial lung imaging, usually resembling that seen in preterm infants with RDS [3,22,26,62,74], evolves throughout the disease and may vary over time, as also happened in our patient [8]. Later in the course of the disease, a nodular and consolidation pattern may be observed [101]. Different lung imaging aspects are expected, as even identical variants of the *ABCA3* gene may present with different phenotypes [118]. The same observation applies to lung histology. All these aspects are described in the literature in association with ABCA3, SFTPB, and SFTPC deficiency [13,15]. Electron microscopy of the lung tissue was reported only in the patient presented by Jackson et al. [67].

Familial history had no relevance for ABCA3 deficiency in the several reviewed cases, as in our family. This is not unexpected, as ABCA3 deficiencies are rare diseases with autosomal recessive inheritance. Most patients were compound heterozygous for the *ABCA3* c.838C>T (p.Arg280Cys, R280C) variant, which, together with the association of various other *ABCA3* variants, may explain the different phenotypes of the subjects (Table 2). Additionally, there is an urgent need for functional studies to quantify the impact of each variant on protein function, expressed as a percentage.

In our case, a compound heterozygous for the *ABCA3* c.838C>T (p.Arg280Cys, R280C) and *ABCA3* c.697C>T (p.Gln233Ter, Q233X, Q233*) variants presented with fatal RDS with neonatal onset. Most probably, a cumulative damaging effect of the c.697C>T (p.Gln233Ter, Q233X, Q233*) variant (in silico predicted as probably pathogenic) significantly contributed to our patient’s severe phenotype.

We support the recommendation that in cases of unexplained, early onset, severe neonatal RDS with persistent radiological and clinical symptoms that are persistent over 1 week, evolving to hypoxemic respiratory failure despite maximal conventional therapy and with transient response to surfactant administration in term and near-term infants, genetic surfactant metabolism dysfunctions should be suspected [6,8,9,13,15,90,91]. NGS offers a crucial role in molecular medicine, a step forward to individualized medicine, as this genetic testing precisely detects single nucleotide variants [73]. NGS, WES, and WGS can help the index case and their family identify the genetic defect and provide genetic counseling. Moreover, extended genetic sequencing may replace the information offered by lung biopsy, an invasive investigation previously recommended in assessing inherited surfactant metabolism disorders [4,6,9,13,74], as a rapid and precise diagnosis is of utmost importance for genetic counseling [10,101]. These techniques may identify VUS that need predictive tools for clarifying the impact on protein function and pathogeny and further genetic counseling [15,73,116].

Our study has several additional limitations. First, the presented patient and his family were investigated by an NGS panel that included a limited number of genes, thus not providing a comprehensive genome analysis. Additionally, we could not perform functional studies to quantify the effect of the *ABCA3* variants, which collectively impacted protein function and led to RDS. Second, we did not assess gene expression or measure the concentration of ABCA3 protein. However, the clinical, paraclinical, imaging, and histological findings, predictive tools, and the results of previously reported studies strongly suggest the pathogenicity of the associated variants in our patient and the need to reclassify these *ABCA3* variants.

## 5. Conclusions

Our study highlighted the necessity, importance, and timing of genetic testing for a step-by-step diagnosis. NGS techniques are considered the gold standard for the genetic investigation of RDS. However, these comprehensive techniques may identify VUS, as in our case, presenting challenges in deciphering their clinical relevance, underscoring the need for continued comprehensive research and review to characterize these variants accurately.

In our case, the patient phenotype, clinical aspects, course, and outcome, along with the literature data and computational predictions from in silico probability tools, indicate that the *ABCA3* c.838C>T (R280C, p.Arg280Cys) and c.697C>T (p.Gln233Ter, Q233X, Q233*) variants had a cumulative effect and should be reclassified as probably pathogenic/pathogenic variants. The data from the extensive review of the literature also support our conclusions.

## Figures and Tables

**Figure 1 biomedicines-12-02390-f001:**
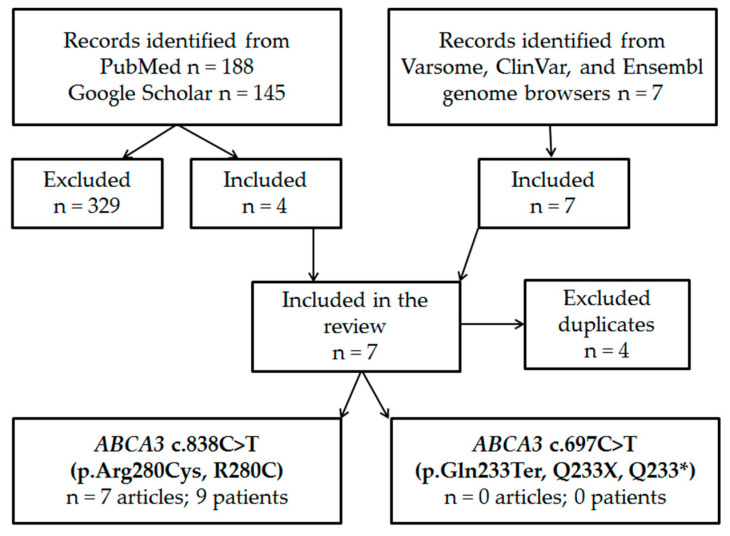
The flow chart of the systematic review on *ABCA3* c.838C>T (p.Arg280Cys, R280C) and c.697C>T (p.Gln233Ter, Q233X, Q233*).

**Figure 2 biomedicines-12-02390-f002:**
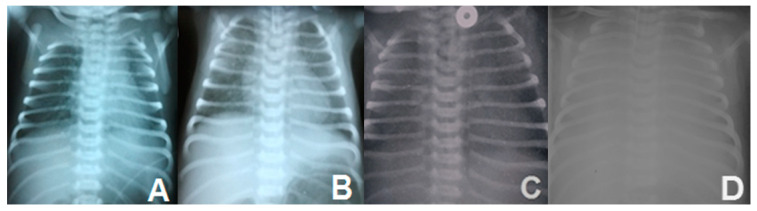
Anterior-posterior thoracic X-rays of the index patient in the supine position. (**A**) DOL 1—small perihilar distelectatic foci. (**B**) DOL 5—reticular, micronodular lung interstitium, confluent along the hilum and in the area corresponding to the superior right lung, air bronchogram. (**C**) DOL 40—diffuse bilateral, symmetrical opacification of the lungs with a granular aspect, extended air bronchogram. (**D**) DOL 60—diminished lung fields, diffuse, symmetrical, almost complete opacification, disappearance of the mediastinum and diaphragm outline (DOL—day of life).

**Figure 3 biomedicines-12-02390-f003:**
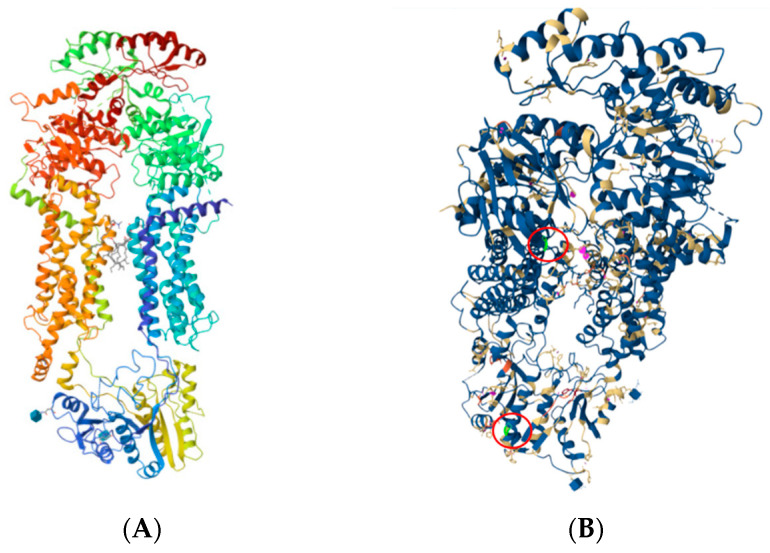
Structure 7W01 of the Cryo-EM structure of nucleotide-free ABCA3. (**A**) A crystal 7W01 structure at a resolution of 3.3 Å. (**B**) The changes are highlighted in green within the 7W01 structure, marked by red circles, and were visualized using Mol* Viewer, a modern web app for 3D visualization and analysis [59].

**Figure 4 biomedicines-12-02390-f004:**
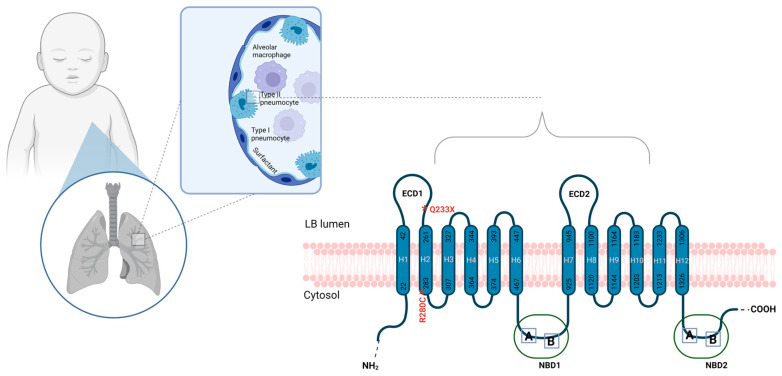
A structural model of the ABCA3 protein, showing the distribution of the compound heterozygous c.838C>T (p.Arg280Cys, R280C) and c.697C>T (p.Gln233Ter, Q233X, Q233*) variants detected in *ABCA3* in this study. The ABCA3 protein comprises 12 putative membrane-spanning helices along the NH2- and COOH- terminal domains, two extracellular domains (ECDs), and two nucleotide-binding domains (NBDs). The critical conserved Walker A and Walker B motifs are indicated as A and B, respectively, within the nucleotide-binding domains. The red asterisk marks the *ABCA3* c.838C>T (p.Arg280Cys, R280C) and c.697C>T (p.Gln233Ter, Q233X, Q233*) variants identified in this study.

**Figure 5 biomedicines-12-02390-f005:**
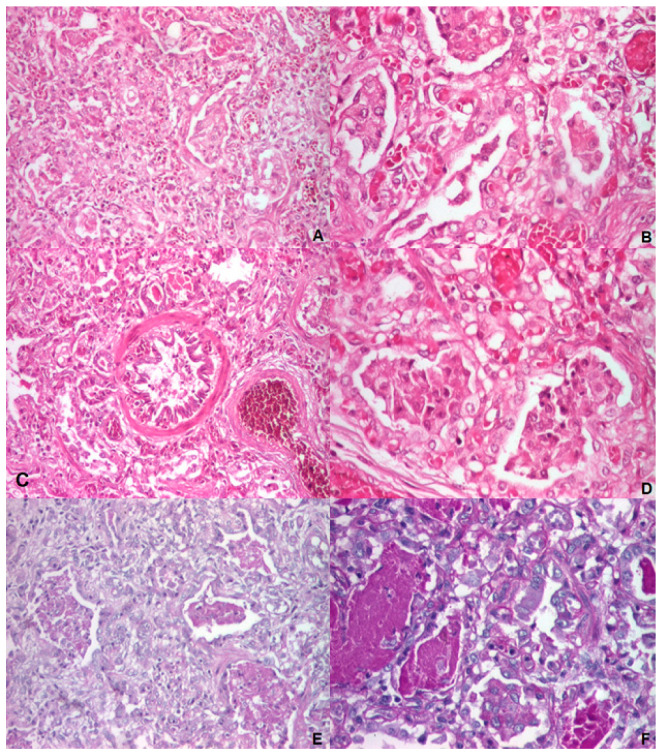
Lung histology—microscopic examination: (**A**) Alveolar proteinosis (×100)—fine, granular, eosinophilic material inside the alveoli, some with a compact aspect; (**B**) thickened alveolar septae and vascular stasis (×200)—additionally, mononuclear cells and fibroblast are present; (**C**) alveoli and bronchiole filled with eosinophilic material (×200)—additionally, atelectatic and emphysematous areas; (**D**) alveoli with pseudostratified AEC II (×200); (**E**,**F**) alveolar proteinosis (×100)—amorphous or granular PAS-positive material inside the alveoli is suggestive of lung alveolar proteinosis; (**A**–**D**) hematoxylin-eosin staining; (**E**,**F**) periodic-acid Schiff (PAS) staining.

**Figure 6 biomedicines-12-02390-f006:**
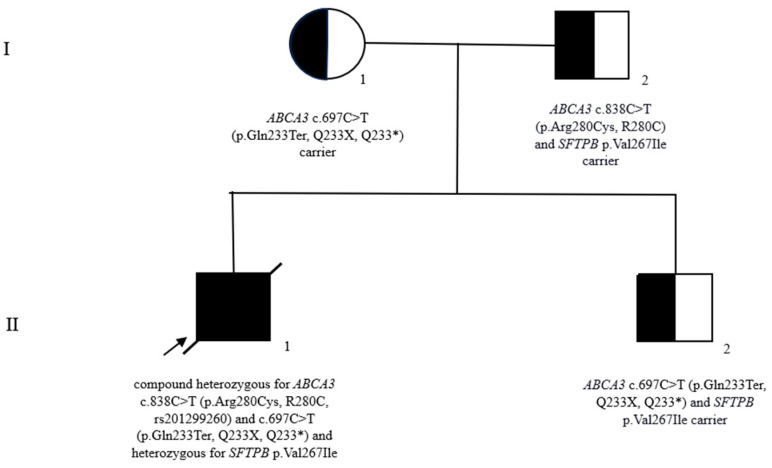
Segregation analysis within the family studied: the squares represent males, and the circles represent females; the arrow points to the index case; the slashed symbols indicate deceased individuals; the black-filled symbols denote individuals with fatal neonatal respiratory distress syndrome; the half-filled symbols represent carriers. Generations are denoted using Roman numerals, while individuals are identified by Arabic numerals.

**Table 1 biomedicines-12-02390-t001:** Consistency analysis of prediction tools for the ABCA3 c.838C>T (p.Arg280Cys, R280C) and c.697C>T (p.Gln233Ter, Q233X) variants.

Variant	Tool	Score/Result	Classification	Comments/Consistency (%)
*ABCA3* c.838C>T (p.Arg280Cys, R280C)	MutPred2	0.543	Possibly deleterious	90%
	PolyPhen-2	0.989	Probably damaging	
	MutPred-LOF	N/A	Not applicable	
*ABCA3* c.697C>T (p.Gln233Ter, Q233X)	MutPred2	N/A	Not applicable (a tool designed for missense mutations)	Not applicable for nonsense variants
	PolyPhen-2	N/A	Not applicable (focuses on missense variants)	Not applicable for nonsense mutations
	MutPred-LOF	0.369	Moderate probability of loss of function	100% consistency in identifying a loss-of-function variant

Note: the consistency percentages (%) indicate the level of agreement among the prediction tools regarding the potential pathogenicity of the variants. N/A—not applicable.

**Table 2 biomedicines-12-02390-t002:** Anthropometric, clinical, radiological, and histological characteristics, as well as the genetic variants, family history, and patient outcomes reported in the literature for the *ABCA3* c.838C>T (p.Arg280Cys, R280C) variant.

Patient Number	Reference	Gender	GA	BW	Respiratory Disease	Imaging Aspects	Lung Histology	EM	*ABCA3* Variant—Allele 1/Parental Origin	Other Associated *ABCA3* Variant(s)/Parental Origin/Allele 2	Variants of Other Surfactant Genes/Parental Origin	Familial History	Outcome	Comments
1.	Somaschini et al., 2007 [62]	Male	39 weeks	2850 g	Severe nRDS (mechanical ventilation)	N/A	N/A	N/A	R280C/wt/	Most probably the second mutation was missed due to restrictive genetic testing protocol	Tested; none reported	No	Died at 2 days	Patient 7 in a case series of 17 cases with inherited deficiency of pulmonary surfactant
2.	Turcu et al., 2013 [63]	Not reported	Term	Not reported	nRDS—Surfactant administration, ventilation, hydroxychloroquine	Interstitial changes on lung CT	CIP	N/A	c.838C>T (p.Arg280Cys)/heterozygous/parental origin not reported	c.2069A>G (p.Glu690Gly)/heterozygous/parental origin not reported	None reported	no	Alive at 13 years	Patient reported in a case series of 323 cases analyzed for inherited deficiency of pulmonary surfactant
3.	Williamson & Wallis, 2014 [64]	Female	Term	Not reported	nRDS—surfactant, mechanical ventilation, multiple corticosteroids courses, oxygen-dependent at 2 years; treated with hydroxychloroquine afterwards	Extensive patchy ground glass opacification and cystic airway changes, predominantly in the upper lobes at 2 years on thoracic CT	DIP at 2 years	N/A	c.838C>T/heterozygous/parental origin not reported	c.2069A>G/heterozygous/parental origin not reported	Tested; no variants discovered	N/A	Final evaluation at 13 years, stable under hydroxychloroquine treatment	Case report
4.	Wambach, 2014 [65]	Female	N/A	N/A	nRDS	N/A	N/A	N/A	Q1589X R280C (homozygous)	Q1589X R280C (homozygous)	N/A	N/A	Died < 3 months	Based on unpublished data
5.		Male	N/A	N/A	nRDS	N/A	N/A	N/A	c.3997_3998delAG (null)	R280C	N/A	N/A	Alive	Based on unpublished data
6.		Female	N/A	N/A	nRDS	N/A	N/A	NA	Q1589X (p.Gln1589) R280C	c.4195G>A (V1399M) (p.Val1399Met)	N/A	N/A	Lung transplantation at 10 months	Based on unpublished data; also reported by Xu et al., 2022 [66]
7.	Jackson et al., 2015 [67]	Female	Term	2870 g	Severe nRDS treated with advanced respiratory support (HFOV, iNO), weaned for respiratory support at 1 year	N/A	Diffuse interstitial fibrosis with alveolar remodeling and prominent AEC II hyperplasia	Small, dense lamellar bodies and occasional fused lamellar bodies	p.R280C (c.838C>T)/paternal (*cis*)	p.Val1399Met (c.4195G>A)/maternal p.Q1589X (C.4765C>T)/paternal (*cis*)	Tested; none discovered	No (parents and one previous sibling without—healthy)	Alive at 3 years; mild speech and motor delay	Case report; p.V1399M is rare and has been previously reported in symptomatic infants [65]; p.Q1589X is predicted to result in a truncated protein [64] Both p.R280C and p.V1399M are predicted to be damaging to ABCA3 protein function by the majority of in silico prediction programs in ANNOVAR [68]
8.	Klay et al., 2020 [69]	Female	Not reported	Not reported	ILD—onset at 19 years with dyspnea; high-resolution chest CT: diffuse ground glass opacities with emphysema located in the apical regions, progressing to severe lung fibrosis	N/A	Diffuse fibrosis with chronic inflammation, unusual interstitial pneumonia, fibroblast foci, or granulomas. Mild AEC II hyperplasia; accumulation of alveolar macrophages	N/A	c.838C>T (p.R280C, rs201299260) (*trans*)/one parent c.875A>T (p.E292V, rs149989682). (*trans*)/the other parent	-	No report on other genetic tests	No	Proposed for lung transplantation at 25 years	Case report; initially diagnosed with drug-induced ILD
9.	Gjeta et al., 2023 [70]	Female	Not reported	Not reported	Onset at 2 years and 7 months with severe pediatric ARDS (requiring invasive respiratory support, prednisolone, and azithromycin treatment), after twoepisodes of upper respiratory tract infections	Chest X-ray showed bilateral opacification, suggesting interstitial bilateral pneumonia. chest CT scan showed bilateral ground-glass opacities	N/A	N/A	c.4261 G>A p. (Gly1421Arg)/heterozygous/paternal	c.838C>T p. (Arg280Cys)/heterozygous/maternal	Tested; none reported	N/A	Alive, after prolonged oxygen therapy and on treatment with oral hydroxychloroquine and fluticasone propionate inhalations	Case report
10.	Ognean et al., 2024 (this study)	Male	37 weeks	2700 g	Severe nRDS—advanced respiratory support (HFOV), prednisone, azithromycin, hydroxychloroquine treatment	Thoracic X-ray—ground glass homogeneous opacity	CPI pattern with lobular remodeling, prominent AEC II hyperplasia, focal PAP pattern, and extensive DIP-like areas; alveolar proteinosis	N/A	p.Arg280Cys (R280C, c.838C>T, rs201299260)/heterozygous/paternal origin	p.Gln233ter (Q233X, Q233*)heterozygous/maternal origin	*SFTPB* p.Val267Ile	No, healthy parents, one healthy sibling despite carrying p.Gln233ter and *SFTPB* p.Val267Ile variants	Died at 77 days of life	Current case report

Legend: GA: gestational age; BW: birth weight; EM: electron microscopy; nRDS: neonatal respiratory distress syndrome; AEC II: alveolar epithelial type II cells; CIP: chronic interstitial pneumonitis; DIP: desquamative interstitial pneumonitis; ARDS: acute respiratory distress syndrome; ILD: interstitial lung disease; PAP: pulmonary alveolar proteinosis; MV: mechanical ventilation; HFOV: high-frequency oscillatory ventilation; iNO: inhaled nitric oxide; CT: computed tomography; ANNOVAR: ANNOtate VARiation (2018Apr16) software tool; N/A: not available.

**Table 3 biomedicines-12-02390-t003:** Pros and cons arguments for *ABCA3* c.838C>T (p.Arg280Cys, R280C) pathogenicity.

	Pros	Cons
Clinical aspects	R280C mutations were reported in association with neonatal RDS and ILD, with the clinical picture and course suggestive of surfactant metabolism dysfunction [62,64,65,67,69,113].	Variable phenotype, usually in association with other mutations, with survival varying between 2 days and over 25 years [69]; severe phenotype was associated in 4 out of the 10 cases reported in the literature (Table 1).
In vitro experiments	R280C mutations result in folding and trafficking defects, increased endoplasmic reticulum stress, and apoptosis induction in lung epithelial cells in vitro [91].	In vitro experiments cannot accurately represent biological functions [86]. Functional impact on ABCA3 protein is less important as compared to other *ABCA3* mutations [69].
DNA analysis	Mutations reported are associated with respiratory disease in the subjects [62,64,65,67,69,113].	Most reported disease-inducing variants are associated with other mutations in *ABCA3* or *SFTPB* genes, with some of these known as pathogenic (ex. Q1589X) [65]; co-occurrence with these mutations may suggest that R280C is a benign variant. Incomplete genetic testing in some patients (for example, Somaschini et al., 2007) [62].
Estimated allele frequency in the population	Rare; suggestive of pathogenicity (surfactant metabolic dysfunction produced by *ABCA3* mutations) [114].	Frequency higher than for other mutations causing ABCA3 deficiency, not offering unequivocal evidence for pathogenicity [60].
Computational predictive tools and conservational analysis	Most indicate a damaging effect of the mutation on ABCA3 protein [60]. Calibrated prediction (examples): mutation assessor—pathogenic moderate; score: 3.515; DANN—pathogenic supporting; score: 0.9994; SIFT4G—pathogenic supporting; score: 0.002, 0.002 [60].	Insufficient power for predicting pathogenicity: Calibrated prediction (examples); SIFT—uncertain; score: 0.002, 0.002; MutationTaster—uncertain; score: 1.1 [60]. Laboratories reporting the variant through submitted clinical-significance assessments without evidence of independent evaluation. Bioinformatic prediction tools do not determine and/or exclude pathogenicity [115,116]. Most of the current prediction algorithms have a 65–80% predictive accuracy for known pathogenic variants and a tendency for low specificity [117]. Predictive bioinformatics seems insufficient for defining the pathogenicity of ABCA3 deficiencies [53,110].

Legend: RDS: neonatal respiratory distress syndrome; ILD: interstitial lung disease; ABCA3 gene: ATP-binding cassette subfamily A member 3 gene; SFTPB gene: surfactant Protein B gene; SIFT:Sorting Intolerant from Tolerant program; DANN: Deleterious Annotation of genetic variants using Neural Networks Computational Tool.

## Data Availability

All data generated or analyzed during this study are included in this article.

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
