# Peer review of "ABCA3 c.838C>T (p.Arg280Cys, R280C) and c.697C>T (p.Gln233Ter, Q233X, Q233*) as Causative Variants for RDS: A Family Case Study and Literature Review"

_biomedicines, 2024, doi:10.3390/biomedicines12102390_

Round 1
Reviewer 1 Report
Comments and Suggestions for Authors
The manuscript titled "ABCA3 c.838C>T (p.Arg280Cys, R280C) and c.697C>T (p.Gln233Ter, Q233X, Q233*) as Causative Variants for RDS: a Review of the Literature and Family Case Study" explores an important topic in clinical genetics. The topic is of clinical relevance, particularly in the context of rare genetic disorders such as neonatal respiratory distress syndrome (RDS). The authors provide a comprehensive review of the literature, which adds value to the manuscript. However, the manuscript requires some revisions before it can be considered for publication.
1. The manuscript lacks detailed descriptions of the predictive tools used for assessing the pathogenicity of the ABCA3 variants. It is recommended to provide more detailed methodology to improve the reproducibility of the study.
2. The statistical analysis and comparison between different prediction tools are insufficient. Including statistical significance levels and consistency analyses would strengthen the manuscript.
3. The discussion on the clinical implications of the ABCA3 variants should be expanded to cover a broader context. Additionally, the manuscript should delve deeper into the molecular mechanisms impacted by these variants.
4. The manuscript would benefit from a thorough language review to enhance clarity and fluency. Some technical terms need to be standardized throughout the text.
5. The manuscript would benefit from the inclusion of additional figures and tables to visually represent the data and literature findings. This could improve the readability and make the results easier to interpret. In addition, ensure that all technical terms are used consistently throughout the manuscript. For instance, the same nomenclature should be used when referring to specific gene variants.
6. While comprehensive, the literature review could benefit from the inclusion of more recent studies to ensure that all relevant data is considered. Additionally, highlighting gaps in the current studies could help justify the importance of your study.
Comments on the Quality of English LanguageThe language of the manuscript should be improved.
Reviewer 2 Report
Comments and Suggestions for Authors
This study aims to clarify the clinical significance of ABCA3 variants found in a specific family case. I consider that the manuscript should have major changes in its presentation. Below I indicate the suggestions that would improve the understanding of the case report: 1. The information presented in the manuscript mixes a case report and review style. It is suggested to unify the presentation of the manuscript as a family case report. 2. The literature review serves as a reference for the interpretation phase of the variants found. From the clinical context, the most important thing is to define the final interpretation of the variants found to provide tools for genetic counseling. 3. It is important to mention that standards and guidelines were considered for the interpretation of the variants (ACMG or other). 4. Include within the presentation of the case the segregation analysis carried out. 5. Ethics committee approval must be mentioned
Round 2
Reviewer 1 Report
Comments and Suggestions for Authors
I now have no further questions; the article is ready for publication.
Comments on the Quality of English LanguageThe quality of the English language in the manuscript is generally clear and comprehensible. However, some areas could benefit from improved clarity and conciseness. Minor grammatical errors and awkward phrasing are present but do not significantly hinder understanding. Overall, with some revisions, the language quality can be enhanced to meet publication standards.
Author Response
"Please see the attachment."

Reviewer 2 Report
Comments and Suggestions for Authors
The adjustments made make the manuscript more understandable.
I recommend accepting the manuscript in the present form.